# Current Concepts in Diagnosis and Management of Patients Undergoing Total Hip Replacement with Concurrent Disorders of Spinopelvic Anatomy: A Narrative Review

**DOI:** 10.3390/medicina59091591

**Published:** 2023-09-03

**Authors:** Richard Ambrus, Pavel Douša, Jozef Almási, Boris Šteňo

**Affiliations:** 1Department of Orthopaedics, Nemocnica Bory Penta Hospitals International, I. Kadlecika 2, Lamac, 841 03 Bratislava, Slovakia; richard.ambrus@pentahospitals.com; 2Department of Orthopaedics and Traumatology, Kralovske Vinohrady University Hospital, 3rd Faculty of Medicine Charles University, FNKV Srobarova 50, 100 34 Prague, Czech Republic; pavel.dousa@fnkv.cz; 3Department of Traumatology and Orthopaedics, Ostrava University Hospital, Faculty of Medicine University of Ostrava, FNO 17. Listopadu Street, Poruba, 708 52 Ostrava, Czech Republic; 4II. University Department of Orthopaedic and Trauma Surgery, University Hospital Bratislava and Faculty of Medicine, Comenius University Bratislava, Antolska 11, Petrzalka, 851 01 Bratislava, Slovakia; boris.steno@fmed.uniba.sk

**Keywords:** spinopelvic, spine stiffness, individualized cup placement, hip spine syndrome

## Abstract

Despite the high success rate of primary total hip replacement (THR), a significant early revision rate remains, which is largely attributed to instability and dislocations. Despite the implants being placed according to the safe zone philosophy of Lewinnek, occurrence of THR dislocation is not an uncommon complication. Large diagnostic and computational model studies have shown variability in patients’ mobility based on the individual anatomic and functional relationship of the hip–pelvis–spine complex. The absolute and relative position of hip replacement components changes throughout motion of the patient’s body. In the case of spinopelvic pathology such as spine stiffness, the system reaches abnormal positional states, as shown with computerized models. The clinical result of such pathologic hip positioning is edge loading, implant impingement, or even joint dislocation. To prevent such complications, surgeons must change the dogma of single correct implant positioning and take into account patients’ individualized anatomy and function. It is essential to broaden the standard diagnostics and their anatomical interpretation, and correct the pre-operative surgical planning. The need for correct and personalized implant placement pushes forward the development and adaptation of novel technologies in THR, such as robotics. In this current concepts narrative review, we simplify the spinopelvic biomechanics and pathoanatomy, the relevant anatomical terminology, and the diagnosis and management algorithms most commonly used today.

## 1. Introduction

Until recently, the standard for primary total hip replacement surgery was based on tried and tested biomechanical assumptions and anatomical landmarks, based on static measurements and imaging, without fully appreciating the dynamic changes and functional positions of the new hip joint and its interplay with movement of the spine, thus affecting the stability and lifespan of the prosthesis. Luckily this has been on a positive trend in the past years with the amount of research on this topic dramatically increasing [1].

While standard preoperative planning usually utilizes only static imaging—in the coronal plane, a standing AP pelvis and a cross-table lateral image of only the hip joint—due to the significance of pelvic tilt and lack of reliable measurements from a single static image, there is a case for using advanced preoperative planning in the sagittal plane with the use of lateral radiographs of the spinopelvic complex in dynamic positions [2,3]. In this way a fully anatomical view of the hip joint is obtained.

Normal functioning of the hip joint requires spinopelvic motion and proper posture [4,5]. The contemporary pioneers of hip–spine disease, most notably Lazennec and then Dorr, have tried to shed light on and simplify the biomechanics as much as possible, defining and standardizing the terminology [4,6,7,8,9,10]. The hip joint can be thought of as but one gear in the clockwork that is the spine–pelvis–hip complex. Each term describes one gear or its motion as they move and change during the functional excursion of an individual. The gears are connected, and some move in tandem while some in opposite directions. The pathology of one these gears puts strain on the whole system, making some work more and some less. Which gear is affected more determines the function and pattern of deterioration of this coordinated movement. When working correctly and in harmony, the anatomical position of the implants is the same as the functional position. The lack of proper spinopelvic junction coordination, however, may negatively affect the functional position of the acetabular component, putting the whole system at risk [4,5].

In addition, the most used guiding principle for acetabular cup positioning in the coronal plane according to the safe zone as described by Lewinnek (LSZ)—inclination and anteversion of 40°/15° ± 10°—has been scrutinized since the landmark paper by Abdel et al., showing the majority of unstable THRs had their components within the LSZ [11,12].

An exemplar group, in which the risk is most prescient, are patients with fused spines undergoing total hip replacement [13]. Numerous studies have shown the risk of dislocation in such patients to be 8–18% and the risk is increased with the number of segments fused [14,15]. However, a large group of patients undergoing THR do not have prior spine surgery but do have a similarly pathologic spine such as sagittal imbalance or stiffness [16]. Heckmann et al. showed 90% of their cohort of late dislocations (>1 year after THR) had spinopelvic imbalance [17,18,19,20,21].

Besides dislocation, the risk of discordance of operative positioning of the implant and a functional acetabular position can lead to less immediate, but clinically nonetheless significant, complications. These include implant impingement, anteriorly—with the cup more retroverted relative to the functional safe zone, for example, during sitting positions, or posteriorly—with the cup more anteverted relative to the functional safe zone, for example, with femoral extension during ambulation or during the recumbent position [4,22,23]. Depending on the individual, this can show as decreased range of motion or even pain during these movements, and more catastrophically, particulate deposition such as metallosis, which leads to osteolysis [24,25]. Suboptimal coverage of the femoral head can also lead to increased contact point pressure and joint reactive force, resulting in edge loading and preferential wear [26]. Increased wear presents clinically as painful implant instability, aseptic loosening, and a revision procedure earlier than expected [27,28].

To understand and address these problems, preoperative planning must take into account functional changes in the hip joint throughout motion made possible by the interplay of the spine, pelvis, and hip [21,29]. These should all be considered to successfully implant the components in the most functional position possible regarding the patients’ specific pathology. There has also been research to address the issue of dysfunctional spinopelvic junction in the preoperative phase. Flexible sagittal spinal deformity is, to a degree, correctable with physiotherapy and manipulation and could play an important role in preoperative patient optimization [30].

Here, we describe and define relevant anatomical terminology and behavior of the spinopelvic complex, and describe the optimal workup of a patient undergoing total hip replacement considering their spinopelvic status.

When first encountered by the practicing hip surgeon, the philosophy of spinopelvic biomechanics with its vast number of lines and angles drawn on radiographs can be overwhelming and confusing.

For a description of a sagittal deformity anteroposterior, images of the pelvis are insufficient and most of the screening measurements are unreliable [2,3,9,31,32]. Lateral radiographs are needed which show the spine from L1 level caudal, the pelvis, hip joints, and proximal femora. To assess mobility of the spinopelvic junction, lateral standing and seated radiographs are used. The deep-flexed position is closer to the functional position of a patient trying to stand up and, when compared to the relaxed-seated position, is more accurate in diagnosing spinal stiffness [33,34,35].

## 2. Nomenclature

The terminology and radiographic measurements are summarized in Table 1 and shown in Figure 1.

Pelvic incidence (PI) is an anatomical parameter of the pelvis which does not change between positions. It represents the anatomic position of the hip joint relative to the sacrum. The higher the PI, the more anterior the hip joint relative to the sacrum. The highest risk of THR instability is in individuals with abnormal anatomy, either too small (<30°) or too high (>65°) PI.

Lumbar lordosis (LL) represents the degree of lumbar position (flexion or extension) as a compensatory mechanism to secure balance of the upper body. Change in LL between standing and sitting is called ΔLL or lumbar flexion angle (LFA). Lumbar spine stiffness is when LFA is less than 20° between standing and deep-flexed positions. A stiff spine is also termed rigid based on the classification used.

When the relationship between PI and LL is not harmonious, clinical function and compensatory capacity deteriorates. Thus, it is called PI-LL mismatch, or unbalanced spine. As a static deformity, it is measured on standing lateral radiographs. Mismatch greater than 10 is termed flatback deformity, and if greater than 20 degrees, it is classified as severe sagittal deformity. In individuals with higher PI, proportionally greater LL is required to maintain balance.

Sacral slope (SS) is the angle of the sacral end plate (S1) and the horizontal line with the patient standing or sitting on an even ground. It is used as a more readily identifiable parameter of pelvic tilt. When SS change (ΔSS) between two postural positions is less than 10°, the individual has significant spine stiffness; the usual comparison is between standing and relaxed or upright seated positions.

Pelvic tilt (PT) has been the most vexing parameter since its definition was different in the arthroplasty and spine literature. Hip surgeons read PT as the tilt of the pelvis measured from the triangle formed by the anterior superior iliac spines (ASIS) and symphysis to the coronal plane, thus calling it the anterior pelvic plane (APP); this is now rather called the anterior pelvic plane tilt (APPt). The spinopelvic tilt (sPT) as previously described in the spine literature as a spinopelvic parameter, namely, the angle formed by the line from the bicoxofemoral axis to the midpoint of S1 and a vertical line. It describes the position of the femoral heads to the base of the spine. A large pelvic tilt represents a higher degree of pelvic retroversion and is a marker of sagittal imbalance.

PI, PT, and SS are codependent parameters, and mathematically could be regarded as PI = SS + PT with, however, SS having higher power. For each 10° of PI, SS increases by 6°–7° and PT by 3°–4° [36].

Acetabular anteinclination (AI) was introduced in spinopelvic research that describes the ventral opening of the face of the acetabular cup in the sagittal plane, such as anteversion, which describes it in the coronal plane. It is measured on lateral radiographs from the long axis of the cup to the horizontal line, such as with measuring inclination on coronal imaging [6].

Pelvic femoral angle (PFA) is the radiographic measurement of angle between the trunk and the thigh. This represents hip flexion relative to the lumbosacral complex. In recent literature, the definitions vary slightly due to the difficulty in assessing the mechanical axis of the femur on a spinopelvic radiograph: Heckmann uses the line from the hip center parallel with the diaphysis, while Grammatopoulos used the 10 cm line to the ventral cortex [21,37]. Smaller PFA in standing position correlates with the degree of fixed flexion deformity of the hip. Change in PFA between positions represents the range of motion (ROM) of the hip joint and can be used to identify the so-called hip users. For every 1° loss of pelvic motion (PT change), PFA must increase by approximately 1° to compensate [19,38]. However, when the pelvis does not tilt, the acetabular cup loses anteversion and cannot accommodate the increasing femoral flexion, placing the patient at risk of impingement.

The combined sagittal index (CSI) is the sum of PFA and AI in the respective standing or sitting positions. It is a recently described parameter simplifying the individual’s functional spinopelvic state and has shown promising numbers in predicting impingement and possible late THR dislocations. An increased standing CSI was suggestive of posterior impingement and anterior dislocation risk, whereas a decreased sitting CSI suggested anterior impingement and risk of posterior dislocation. Factors contributing to abnormal CSI and consequently risk of dislocation are spine stiffness, low PI, and increased PFA relative to AI. Grammatopoulos et al. applied CSI in their cohort and showed significantly increased dislocation risk in patients with standing CSI outside of the range 205° to 245°. In patients with sagittal spinal imbalance, the range is narrower, i.e., 215° to 235° [37].

Mobility of the spinopelvic complex is assessed using the change in one of these parameters between functional positions. The change in AI can be measured postoperatively. Preoperatively, as it moves based on pelvic rotation, change in PT or SS is used. Spinopelvic stiffness can be described using either ΔSS or ΔLL/LFA. LFA is more appropriate to use when using flexed-seated radiographs as ΔSS is likely overestimated [17,33].

Stiffness is then qualitatively described in the literature as stuck standing (anterior pelvic tilt) and stuck sitting (posterior pelvic tilt), depending on the PT or APPt in which the spinopelvic unit is fixed.

In normal spinopelvic kinematics, when moving from standing to sitting, the pelvis rotates posteriorly, thus increasing PT by about 20°, increasing AI by 15°–20°. This allows the needed 55° to 70° change in PFA for an upright sitting position with the trunk–thigh angle being approximately 90° [20,39,40]. Loss of pelvic motion requires an increase in femoral motion to reach the same functional position. This is the main reason why implanting based on anatomical landmarks and in an anatomical position, using the Lewinnek safe zone, while operating on a patient in a supine or lateral decubitus position, without considering the spinopelvic deformity and mobility, may result later in abnormal implant interaction.

## 3. Management Algorithms

With proper notation and identification of needed spinopelvic anatomy in its functional states, the question remains how to deal with this information and how to adjust the component positioning. The hypothesis is that component position needs to be shifted from the anatomical position, in the sagittal plane, to allow impingement-free ROM in the presence of spinopelvic pathology, as then the spinopelvic unit behaves differently. On the femoral side, this means anatomic implantation, restoring medial offset, length, and neck torsion. Modern prosthesis types used with the best survival tend to be non-modular and cementless, with a trapezoidal shape. This means that adjustments rely heavily on the side of the acetabulum. This is also reinforced by the insufficiency of cup position adjustment during functional movement in patients with spine stiffness [10,12,41].

Throughout recent years, many different classifications and suggested treatment algorithms have been proposed. The treatment algorithms developed look for either high-risk patients based on proven risk factors of sagittal imbalance or try to provide a personalized implant position. All look for instances where standard anatomical positioning of the implants might result in suboptimal results based on pathological functional positioning.

One of the original treatment algorithms was based on the modified Bordeaux classification of spine–hip relationships (SHRs), stemming from spine surgery research of sagittal spinal deformity [42,43]. Based on standing and relaxed seated lateral radiographs, lumbopelvic complex (LPC) types (1—hip users or 2—spine users) based on PI (pathoanatomy) and imbalance with respect to SS change (mobility), PI-LL mismatch (deformity), and sagittal vertical axis (SVA), patients were divided into classes A–D. Their adjustments of the cup position are from the anatomical cup position based on the transverse acetabular ligament (TAL) and anatomic landmarks, shifting the cup version by 3.5° for every 10° lack of PT change from standing to sitting. This algorithm notably differentiates hip–spine syndrome as fixed flexion contracture which tends to resolve after THR is placed in the anatomical position, in contrast to spine–hip syndrome which is a disease of the spine, and then affects THR.

Luthringer and Vigdorchik’s Hospital for Special Surgery (HSS) hip–spine workgroup’s classification and algorithm is arguably the most simplistic and utilizes already commonly used and quickly adaptable principles [44,45]. They describe four categories of hip–spine relationships based on alignment and mobility using relaxed seated radiographs. Further adjustment of group 2 (stiff spines) was based on the direction of stiffness measuring the standing APPt. Their acetabular cup position targets are with respect to the functional pelvic plane/coronal plane and not local anatomical landmarks such as TAL, adjusting by increments of 5°. This makes it compatible with C-arm or navigation-assisted cup implantation, while also being easy to adapt by hip surgeons using the freehand technique.

The Dorr classification combined most contemporary knowledge into quite a detailed workup of not just functional acetabular positioning by itself, which is also dependent on spinopelvic morphology and mobility, but also based on the distal components of femoral mobility and proximal femoral anatomy, presenting the groups as a continuous spectrum [36]. Their group also put forward several simplified nomograms. The desired acetabular component position could be calculated based on the phenotype of sagittal imbalance, and mathematically determined using the formula for the sacro-acetabular angle (SAA) and sacral slope (AI = SAA−SS_standing_). This is reciprocally adjusted based on the femoral version to maintain an optimal combined anteversion of the whole unit.

Tezuka, and later Grammatopoulos, developed the combined sagittal index (CSI) as a combined implant measure based on PFA using an AI for postoperative dislocation risk prediction [37,46,47]. The index was also extended for preoperative planning; the desired AI is based on the standing spinopelvic radiograph, and detects pathologic anatomical changes and deformities. The observation is that standing PFA does not change significantly after THR, but increases by 3 degrees. The AI during surgery is based on PFA while targeting a standing CSI between 205 and 245 degrees. If stiffness or other risk factors are present, the interval is narrower. The risk factors mentioned in these studies were low PI, highly retroverted pelvis in standing position, and flatback deformity. The CSI concept is a good determinant of functional safe zone limits to avoid dislocation and instability. Furthermore, by targeting the middle of this safe zone, one could maximize the chances of reaching the target position and provide the patient with optimal ROM. Their group also applied the hip-user index, verifying that high-risk patients with HUI > 80% would be classified with stiff and unbalanced spines. Mobility was assessed with deep flexed radiographs and ΔLL.

All algorithms have the same basis: identify sagittal deformity and loss of pelvic motion, where the standard anatomical position is different to the functional position. This offers four main distinct categories. Some algorithms, however, go deeper and provide mathematical equations for personalized targets or nomograms. The key step is to identify the high-risk patient. Dangerous anatomy such as low PI, spine stiffness, flatback deformity, and excessive standing retroversion of the pelvis are the most prevalent risk factors for dislocation [17,41,45]. Imbalance is identified on standing lateral images while mobility is better assessed comparing standing to deep-flexed sitting radiographs and using lumbar flexion [34].

Aside from sagittal implant position planning, surgeons need to consider other things. Picking the surgical approach based on the direction of expected instability risk, such as direct anterior or posterior in patients with expected anterior or posterior impingement based on CSI, can be protective against dislocation [37]. The next question regards choosing implant types with larger diameter heads, to adjust the femoral version for the target combined anteversion, or choosing a high offset implant for increasing impingement-free ROM [48]. Dual mobility implants protect against the most severe adverse spinopelvic mobility instances and should be useful in the high-risk patient [49,50].

However, picking a single algorithm and implanting the prosthesis according to the plan in the operating theatre remains difficult. After classifying the patient, identifying the risk factors, and calculating the target cup position, the surgeon then needs the technical support and know-how to achieve these non-standard targets. With the freehand technique, proprietary jigs are usually made for positioning of the cup in 40° coronal inclination and 20° anteversion, and the surgeon’s accuracy is within 5 degrees using coronal plane and OR landmarks [51]. For precise individualized cup placement, technology should be very helpful [52,53]. This technology, however, is hard to integrate into a normal workflow, especially for higher volume surgeons.

The future calls for the use of the most easily adaptable and simple method while securing accuracy and reproducibility. However, ethical considerations arise when regarding, for example, adaptation of technologies with higher degrees of autonomy or inclusion of artificial intelligence. Regulatory, ethical, and legal frameworks also often lag behind the rapid advancement of technology. Most surgical robots today, however, are still under the direct control of the practicing physician, with whom the responsibility and decision making still lies [54].

## 4. Conclusions

The topic of abnormal spinopelvic anatomy and functional status with relevance to the practicing hip surgeon has entered mainstream, with the number of diagnostic, computational, and anatomical studies ever increasing. Early application of this knowledge resulted in the adaptation of several management algorithms. Similarities between them include detection of the high-risk patient with regards to pelvic incidence, sagittal spinal deformity, and spine stiffness. This narrative review compiled the current concepts used, placing emphasis on implementing the spinopelvic preoperative radiographic planning as a standard. When adapted by the surgeon, it should guide their surgical technique to account for the functional positions of the awake and ambulatory patient, the implant choice, or the surgical approach. Future research and mid- to late-term follow-up of clinical studies will show the significance of benefits gained with individualized cup positioning.

## 5. Case Example

Using the radiographs from Figure 1, this 73-year-old lady with severe right-sided osteoarthritis shows PI at 65°, on the upper end of the normal range. Her LL is harmonious with the PI despite significant osteophytosis, within 10° of PI, PI-LL = 8°; therefore, normal spinal alignment and balanced spine are indicated. In standing position, her pelvis is slightly anteverted, PT = 15°. Standing PFA = 181°. When shifting to sitting position, her spinopelvic unit adapts very little. ΔSS and ΔPT show just 1° of change and her lumbar lordosis adapts slightly, but LFA is only 13°, describing a stiff spine. Her sitting PFA is 117° for ΔPFA of 64°. Measurements made using semi-automated computer software Surgimap^®^ Spine (Nemaris Inc.^TM^, NY, USA) validated the sagittal balance measurements [55,56].

Using the 2018 HSS hip–spine classification [44], she is group 1B; as her pelvis is stiff in anterior pelvic tilt, she is “stuck standing stiff.” She benefits from increased coronal inclination and anteversion, of 45° and 25°–30°, respectively, according to their algorithm.

Surgery was performed in standard fashion, via the direct anterior approach with the patient supine on a flat table using the freehand technique implemented by a high-volume surgeon experienced with this approach. During cup placement, the component was first placed according to the standard target of inclination and anteversion 40° and 15°, then the rod was adjusted only in sagittal plane, estimating operative anteversion 25°. The rest of the surgery was performed in routine fashion, and femoral version was constitutional, estimated around 15°.

No adverse events occurred, and satisfactory mobility was achieved postoperatively as reported by the patient at the latest follow-up. Repeated radiographic assessment was made.

In Figure 2, standard coronal measurements on standing AP pelvis showed the cup being in 44° of inclination and 31° of anteversion (measured using the Lewinnek method). On sagittal measurements, standing AI = 36° and sitting AI = 53°, without indication of impingement. Her pelvis rotated posteriorly slightly with PT_standing_ = 20° and little mobility was returned—ΔSS = 8° and ΔPT = 9°—although this was still classified as stiff. This, along with the expected surgical error, could account for the change between estimated anteversion and resultant anteversion of the acetabular component.

According to CSI, her post-THR PFA_sitting_ = 134° and PFA_standing_ = 188°. With the safe range of CSI_standing_ being standard, at 205° to 245°, her resulting value of CSI_standing_ = 224° is safely in the middle ranges of the safe zone, indicating an expected low risk of late dislocation.

## Figures and Tables

**Figure 1 medicina-59-01591-f001:**
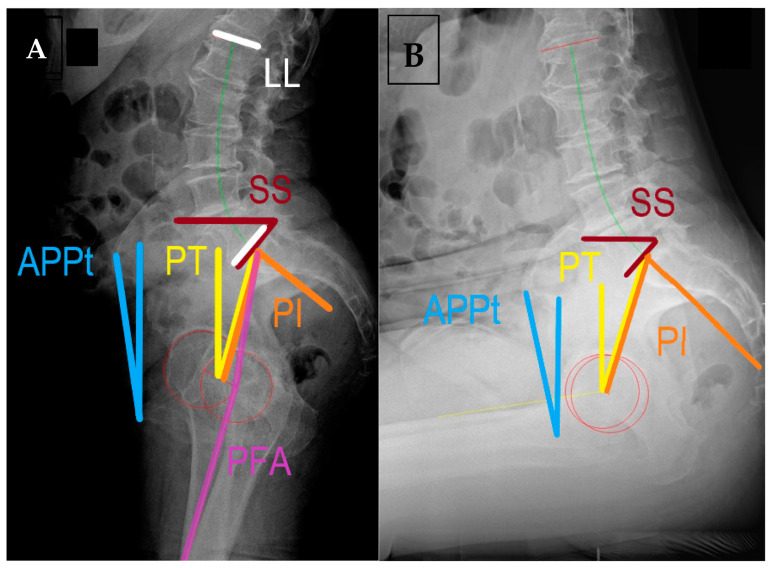
Lateral spinopelvic radiographs of a patient without spinal deformity with spine stiffness and anteverted pelvis (“stuck standing”) in (**A**) standing and (**B**) sitting positions with the commonly used measurements. APPt = anterior pelvic plane tilt, LL = lumbar lordosis, PI = pelvic incidence, PT = spinopelvic tilt, PFA = pelvic femoral angle, SS = sacral slope. Measured using SurgiMap^®^ (Nemaris Inc.^TM^, New York, NY, USA).

**Figure 2 medicina-59-01591-f002:**
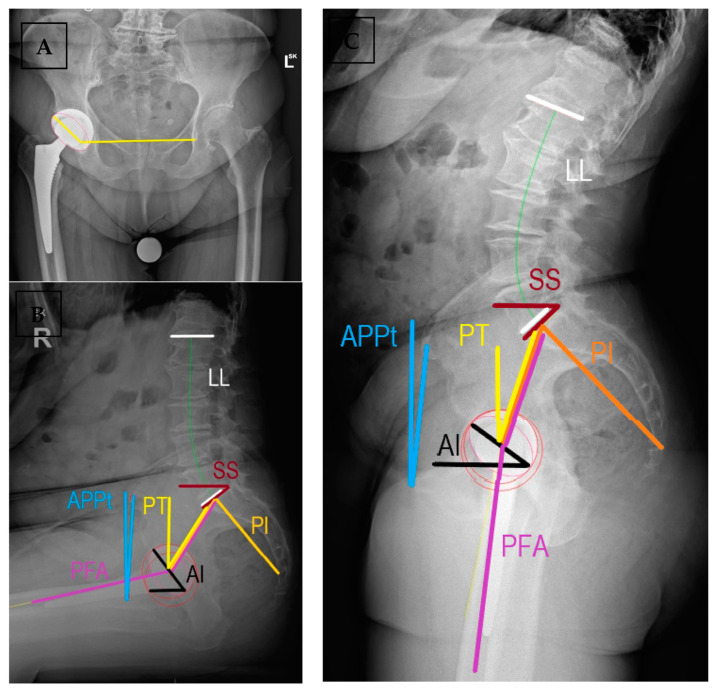
Radiographic assessment of spinopelvic parameters and implant positions at early follow-up after THR. (**A**) Coronally, acetabular inclination and anteversion, and (**B**) sagittal parameters of implant positioning and spinopelvic parameters are shown in sitting and (**C**) standing positions. AI = acetabular anteinclination, APPt = anterior pelvic plane tilt, LL = lumbar lordosis, PI = pelvic incidence, PT = spinopelvic tilt, PFA = pelvic femoral angle, SS = sacral slope. Measured using SurgiMap^®^ (Nemaris Inc.^TM^, NY, US).

**Table 1 medicina-59-01591-t001:** Summary of commonly used terms describing sagittal spinopelvic pathologies and hip-spine syndrome.

Term	Definition	Relevance	Normal Values
**Pelvic incidence (PI)**	Angle between the perpendicular to the midpoint of sacral plate (S1) and the line connecting it to the center of the bicoxofemoral axis	Represents the relative anatomic position of the hip joint to the sacrum.	40°–65°
**Lumbar lordosis angle (LL)**	Angle between the superior plate of L1 to the sacral end plate (S1)	Compensatory to pelvic morphology and position.	Within 10° of PI
**Sacral slope (SS)**	Angle of the sacral end plate (S1) and the horizontal line	Preferred parameter to assess spinopelvic motion, related to PI and PT	SS_standing_ > 30° OR 0.75 × PI SS_sitting_ 5°–30°
**Anterior pelvic plane tilt (APPt)**	Functional pelvic plane as a triangle formed by ASIS and pubic symphysis relative to the vertical line	Used for pelvic tilt in arthroplasty literature, describes the rotation of pelvis in the sagittal plane	0 or slightly anteverted in standing retroverted in sitting
**Pelvic tilt (PT)**	Angle formed by the line from bicoxofemoral axis to the midpoint of S1 and a vertical line	It describes the position of the femoral heads to the base of the spine, related to SS and PI	PT_standing_ < 22° ΔPT ≈ 20° PI = SS + PT
**Pelvic femoral angle (PFA)**	Angle between the line connecting midpoint of S1 endplate with the center of the measured femoral head and femoral mechanical axis	Assesses flexion deformity and femoral motion. Does not change post-THR (≈3°)	ΔPFA 55°–75° PFA_standing_ 180–190 PFA_sitting_ 120–130 Proportionality with PI
**Acetabular anteinclination (AI)**	Angle between the long axis of the cup and the horizontal on lateral radiographs	Sagittal plane orientation of the acetabular cup, represents anteversion	AI_standing_ 25°–45° AI_sitting_ 45°–65° Surgeon dependent
**PI-LL mismatch**	Difference between PI and LL angle	Compensatory ability of lumbosacral spine to changes in pelvic tilt; sagittal balance	<10 in standing lateral radiographs
**Combined sagittal index (CSI)**	CSI = PFA + AI	Validated predictor for acute and late dislocations in postoperative assessment. Possible to plan AI based on PFA	CSI_standing_ 205°–245° If low PI, sagittal imbalance, stiffness: range 215°–235°

## Data Availability

Data sharing not applicable.

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
