# Peer review of "Current Concepts in Diagnosis and Management of Patients Undergoing Total Hip Replacement with Concurrent Disorders of Spinopelvic Anatomy: A Narrative Review"

_medicina, 2023, doi:10.3390/medicina59091591_

Round 1
Reviewer 1 Report
Specific Comments
Title: “spinopelvic anatomy” please change to “spinopelvic function”.
Please include at the end of the title the type of study: A narrative review.
1. Introduction
Line 61. Please include some sentences in this paragraph explaining the next ideas:
If possible, physiotherapy for spine sagittal imbalance or reversible stiffness is recommended in order to get the more physiological values as possible before surgery. Spine sagittal imbalance or degenerative stiffness are not static disfunctions, they are not anatomical, they are functional, and can change. Thus, it is better to have them as close as possible to the physiology before surgery, and maintain them after.
One recommended citation: “Barbosa AC, Martins FL, Barbosa MC, Dos Santos RT. Manipulation and selective exercises decrease pelvic anteversion and low-back pain: a pilot study. J Back Musculoskelet Rehabil. 2013;26(1):33-6. doi: 10.3233/BMR-2012-0347. PMID: 23411646.”
3. Nomenclature
Line 98. “of lumbar motion”. Maybe it is better to correct to “lumbar position”, due to the fact that LL represent a position and not a range of movement.
Figure 1. Is it possible to add the lines for AI or include them into another figure?
Is it possible to include some values about reliability of the measures using SurgiMap® (Nemaris Inc.TM, 161 NY, US)?
4. Management algorithms
Line 196. Please, explain the acronym SVA.
Line 260. Please explain a little more this concept: “anatomical 40-20 positioning”
6. Case example
Figure 2. Please include reference letters A or B in the images.
Line 292. Please clarify a little more in the figure: “standard coronal measurements on standing AP pelvis showed cup sitting in 44° of inclination and 31° of anteversion (measured using Lewinnek method)” Is it on standing or in sitting?
Author Response
Please see the attachment.
Description of the article was changed to narrative review.
Title was left with the wording as is, as this reflects also the target audience for the special issue of clinical anatomy.
The reminder of the research on non-operative management of spinopelvic imbalance was welcomed and included.
In the section “Nomenclature” wording was fixed to “lumbar position.”
Acetabular anteinclination (AI) of the native acetabula is usually measured on cross-table (or `true`) lateral image of the hip which is part of the preoperative workup but only as a static estimated measurement. True anatomical anteversion in a proven fashion is obtained only via a CT scan. Neither address the functional state. AI itself as a spinopelvic unit is defined as the sagittal position of the implant.
The validity and reliability of the used measurement computer software was added. Due to stylistic reasons, it landed in the section “Case example.”
The acronym SVA was defined – sagittal vertical axis.
The wording for the colloquial term of “anatomic 40/20 positioning” was changed and better explained.
Letter markings were added to all figures.
In the line 292 of the previous version, the misleading usage of sitting as an idiom rather than as a position of a body was changed to be clearer. The measurement in that sentence talked about a standing position of the body.

Reviewer 2 Report
I have reviewed the research article titled "Current concepts in diagnosis and management of patients undergoing total hip replacement with concurrent disorders of spinopelvic anatomy" by Ambrus et al., submitted to Medicina. This study delves into a critical aspect of total hip replacement (THR) involving patients with coexisting spinopelvic anatomy disorders. While the topic is undoubtedly important, there are several key aspects that need significant improvements to elevate the quality and influence of the article. I would strongly recommend a major revision of the manuscript to address these concerns.
The structure of the abstract requires clarity enhancement. The current abstract lacks a distinct structure, making it somewhat challenging for readers to comprehend the main points effectively. I would suggest restructuring the abstract into well-defined sections that succinctly introduce the problem, lay out the objectives, outline the methodology, summarize the findings, and discuss the potential implications of the study's outcomes. This restructuring would considerably improve the abstract's readability and its capacity to convey its key messages effectively.
Moreover, the manuscript could benefit from a more comprehensive literature review. Presenting a concise yet well-rounded overview of the existing literature on spinopelvic anatomy, THR, and associated complications would help set the context and establish the significance of the study's goals. This addition would not only highlight the gaps in current knowledge that this research aims to address but also enhance the abstract's credibility.
While the abstract touches on the significance of personalized implant positioning, it falls short of explicitly discussing the clinical relevance of the findings. It would be beneficial to expand on how the proposed concepts and management strategies translate into improved patient outcomes, a reduction in complications, and an overall enhancement in the quality of life for individuals undergoing THR with concurrent spinopelvic disorders.
Incorporating ethical considerations is also essential. Since the study suggests personalized implant positioning and emerging technologies like robotics, it is imperative to address the ethical implications. It would be wise to touch upon potential challenges, risks, and benefits associated with these approaches, ensuring that the safety and well-being of patients are paramount.
The paper introduces complex concepts such as spinopelvic biomechanics and pathoanatomy, yet some terms lack clear definitions. To enhance clarity and accessibility, I recommend providing succinct explanations of these key terms and concepts.
Finally, concluding the paper with a section that discusses the broader implications of the study's findings for clinical practice, potential advancements in THR techniques, and the possible future directions in the field of spinopelvic disorders and hip replacement would provide a satisfying closure.
Addressing these concerns through a major revision will undoubtedly enhance the article's value and make it a significant contribution to the field of orthopedics. It will also help improve patient care in total hip replacement surgeries by addressing the complexities of concurrent spinopelvic disorders.
ok
Author Response
Please see the attachment.
Abstract was rewritten into more clear sections. The recommended structure using sections such as objectives, methodology, discussion was not entirely appropriate for this format. Instead, the structure relies on the talking points further expanded on in the body of the text.
Explanation of the relevant anatomy and biomechanics was dispersed too wide in the previous version. The new version provides a concise section talking about the normal biomechanics of the area.
The relevant anatomy is defined and explained in the section terminology and the table provides also the researched, definite normal values of said anatomical and functional radiological parameters.
Clinical relevance was expanded into a separate paragraph. Mentioned here are the implications of ignoring the current research and spinopelvic status assessment.
Added was a section discussing the broader implications of adaptation of these concepts as well as ethical considerations of using emerging technologies. Benefits have been mentioned throughout the text, for example around the line 250.
Key terms describing the hip spine relationships are defined in the paragraph about biomechanics and also in terminology section. The terms are grouped into one Table in this section.
The last paragraphs of the body were expanded addressing the clinical implications of adaptation of such management algorithms into practice.
A separate conclusion was added.

Round 2
Reviewer 1 Report
I want to thank the authors the implementation of almost all the suggestions.
Reviewer 2 Report
The manuscript may be accepted in present form.